# VFXMASTER: UNLOCKING DYNAMIC VISUAL EFFECT GENERATION VIA IN-CONTEXT LEARNING

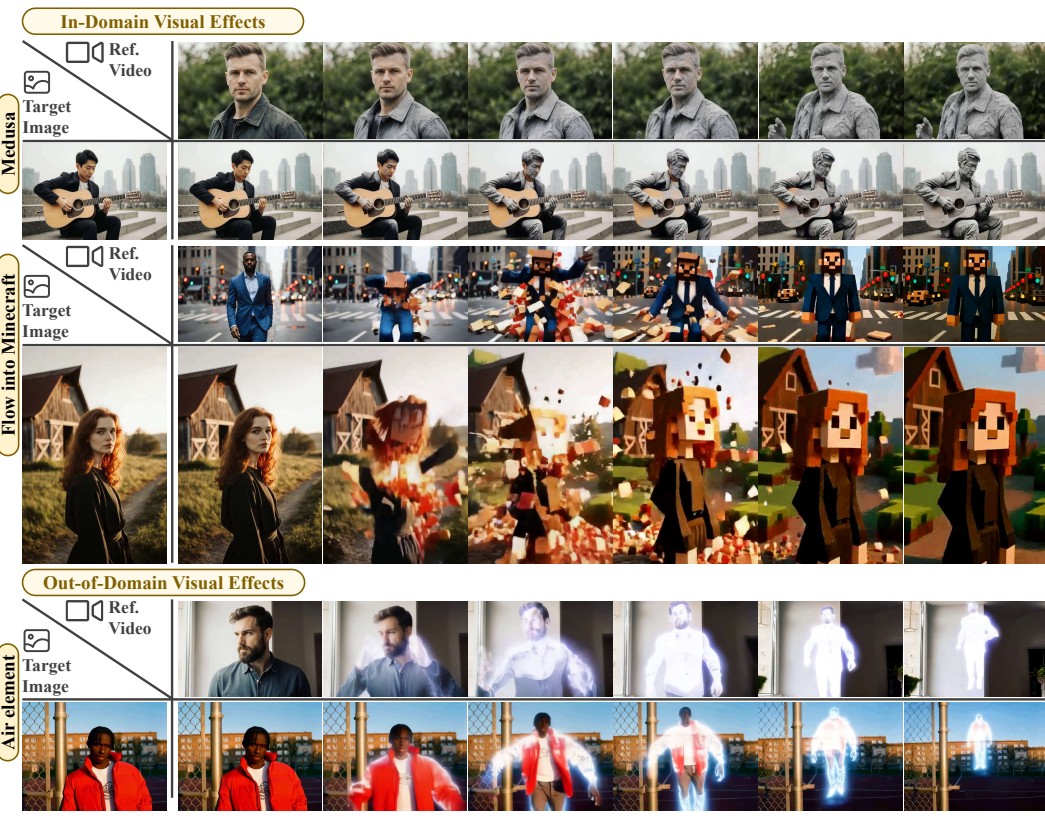

Figure 1: **VFXMaster** is a unified reference-based cinematic visual effect (VFX) generation framework that can reproduce the intricate dynamics and transformations from a reference video onto a user-provided image. It not only shows outstanding performance on in-domain effects, but also strong generalization capability on out-of-domain effects.

## ABSTRACT

Visual effects (VFX) are crucial for the expressive power of digital media, yet their creation remains a major challenge for generative AI. Prevailing methods often rely on the one-LoRA-per-effect paradigm, which is resource-intensive and fundamentally incapable of generalizing to unseen effects, thus limiting scalability and creation. To address this challenge, we introduce VFXMaster, the first unified, reference-based framework for VFX video generation. It recasts effect generation as an imitation task, enabling it to reproduce diverse dynamic effects from a reference video onto a target content. Critically, it demonstrates remarkable generalization to unseen effect categories. Specifically, we design an in-context conditioning strategy that prompts the model with a reference example. We use an in-context attention mask to precisely decouple and inject the essential effect attributes, allowing a single unified model to master the effect imitation without information leakage. In addition, we propose an efficient one-shot effect adaptation mechanism to boost generalization capability on tough unseen effects from a single user-provided video rapidly. Extensive experiments demonstrate that our method effectively imitates various categories of effect information and exhibits outstanding generalization to out-of-domain effects. To foster future research, we will release our code, models, and a comprehensive dataset to the community.

# 1 INTRODUCTION

Visual effects (VFX) are an integral component of modern digital media, greatly enriching the visual expressiveness of films, games, and social media content. Traditional VFX production is a time-consuming and labor-intensive process that demands specialized skills across multiple stages, including modeling, rigging, animation, rendering, and compositing (Du et al., 2021). Recent and rapid advancements in generative AI bring revolutionary opportunities for content creation (Ma et al., 2025; Wang et al., 2024). In particular, the growing maturity of video generation models (Yang et al., 2024; Kong et al., 2024; Wan et al., 2025; HaCohen et al., 2024) is ushering in a new era of controllable content synthesis. However, due to data scarcity and sophisticated transformations, the dynamic visual effect generation task is still rarely studied so far.

Existing video generation models, pretrained on large-scale real-world datasets, possess powerful content generation capability. However, VFX often contain anti-physical, surreal, and counter-intuitive elements, such as the particle dynamics of an energy beam or the brilliant patterns of magical elements (Bai et al., 2025b). These highly abstract and imaginative concepts represent an out-of-domain challenge that falls significantly outside the knowledge scope of pretrained models. Even with highly detailed text prompts, these models struggle to produce the desired effects accurately. Furthermore, prevailing controllable generation methods focus on spatial-aligned conditions, such as keypoints (Gu et al., 2025; Jeong et al., 2025), depth maps (Peng et al., 2024; Wang et al., 2025), or edge sketches (Yang et al., 2025b; Geng et al., 2025), which cannot effectively model the intricate, unstructured dynamics and textures of visual effects. Several recent works have achieved preliminary visual effect generation by finetuning Low-Rank Adapters (LoRA) on pretrained models (Hu et al., 2022; Liu et al., 2025).

However, the one-LoRA-per-effect paradigm suffers from a fundamental scalability bottleneck. This paradigm requires dedicated data and training for each effect. More critically, this closed-set training paradigm strictly confines the model's capability to known effects. It is unable to handle any unseen effect category, which severely limits the system's applicability and the user's creative freedom. Recently, Mao et al. (2025) has made initial attempts using the LoRA-MoE architecture for learning the effects in the training set jointly, but they still cannot generalize to unseen effects. So how can we break through this limitation and achieve straightforward VFX video generation? We observe that videos sharing the same VFX differ only in subjects and backgrounds, but maintain similar dynamics and transformation process. This observation inspires us to regard two videos with the same VFX as a reference-target data pair for in-context learning, i.e., using one video as reference to guide the model in reproducing its visual effect. Such a reference-based paradigm maximizes data utilization and enables a unified framework for learning a general VFX imitation capability rather than memorizing specific effects. This provides users with an intuitive and friendly creative tool.

In this work, we propose VFXMaster, the first unified framework for VFX video generation. By learning from reference effects via in-context learning, VFXMaster integrates diverse effects into a single model and demonstrates strong generalization capability beyond its training set. Specifically, we design an in-context learning paradigm where a reference prompt-video pair serves as an example, while a target prompt and the first frame act as a query to condition the model for the target video. However, the reference context contains components irrelevant to the effect. To prevent information leakage and interference, we introduce an in-context attention mask mechanism to learn only the visual effect from the reference example. Furthermore, to enhance generalization to Out-of-Domain (OOD) effects, we design an efficient one-shot effect adaptation strategy that introduces a set of learnable concept-enhance tokens to further learn the fine-grained VFX dynamics and transformations from a single user-provided sample. With a low-cost token finetuning, the model can rapidly improve the generalization capability on tough OOD samples.

We conduct extensive experiments on existing benchmarks to evaluate our method. In addition, to validate generalization capability, we build a new OOD test set and design a comprehensive evaluation metric tailored for reference-based effect generation. The results demonstrate that VFXMaster achieves remarkable VFX generation performance and strong generalization capability when facing OOD data. To support future research, the curated dataset and designed metric will be released. In summary, our contributions are as follows:

- We propose VFXMaster, the first unified reference-based framework for VFX video generation. It achieves high-quality effect imitation and strong generalization to unseen effects.

- We introduce an in-context conditioning strategy that incentivizes the model to reproduce the visual effect from a reference example onto a target image. We design an in-context attention mask to focus on the visual effect and prevent information leakage.

- We propose an efficient one-shot effect adaptation strategy. Using a set of concept-enhance tokens enables the model to further learn fine-grained VFX from a single video, significantly improving its generalization capability for tough OOD scenarios.

## 2    RELATED WORK

### 2.1    CONTROLLABLE VIDEO GENERATION

Diffusion models have significantly advanced video generation, as evidenced by the work of (Ho et al., 2020; Song et al., 2020a;b; Rombach et al., 2022), facilitating numerous innovative methodologies. Among these, the Diffusion Transformer (DiT) (Peebles & Xie, 2023) leverages Transformer architectures to effectively capture long-range dependencies, thereby improving temporal coherence and dynamics. Based on DiT, CogVideoX (Yang et al., 2024) utilizes 3D full attention to ensure spatial–temporal consistency, whereas Hunyuan-DiT (Kong et al., 2024) integrates large-scale pre-trained models to enhance contextual details. Controllable video generation has also garnered considerable interest for its applications in video editing and content creation. Several studies (Bai et al., 2025a; 2024) introduce 3D control signals to manipulate object positions, motion trajectories, and camera perspectives within the 3D scene. Other work (Yang et al., 2025a)incorporates VLM as a motion planner to generate physically plausible videos, or by introducing additional mechanisms such as StyleMaster (Ye et al., 2025), which combines style extraction mechanism with motion control to enhance video stylization and transfer. In addition, ControlNet (Zhang et al., 2023) facilitates image generation through control signals by replicating designated layers from pre-trained models and connecting them with zero convolutions. FlexiAct (Zhang et al., 2025) utilizes the denoising process's capability to focus on various frequency components over time, facilitating the transfer of motion from a reference video to a selected target image. Beyond controllability, other works extend video generation capability. Wan-FLFV (Wan et al., 2025) generates smooth transitions between user-specified starting and ending frames, while VACE (Jiang et al., 2025) integrates ID-to-video generation, video-to-video editing, and mask-based editing into a unified model, enabling efficient video generation and editing.

### 2.2    VISUAL EFFECTS GENERATION

Visual effects (VFX) have recently been explored through video generation models, providing a more efficient alternative to traditional production. Despite advancements in general video generation, the generation of controllable visual effects (VFX) remains insufficiently explored, largely due to the lack of VFX data and the constraints of conditional control. MagicVFX (Guo et al., 2024) is restricted to adding green-screen overlays, lacking extensibility and controllability. VFXCreator (Liu et al., 2025) generates effects by training a separate LoRA for each case, which limits it to single-effect video generation. Although OminiEffects (Mao et al., 2025) represents a step forward by employing LoRA-MoE to enable spatially controllable composite effects, the supported effect types are still narrow and confined to in-domain combinations. Despite these advances, current approaches cannot unify diverse effects within a single framework and show limited generalization to out-of-domain effects. In this work, we propose the first unified framework for VFX video generation to fill the gap in previous research, offering a comprehensive solution for this task.

## 3    METHOD

Controllable visual effect (VFX) generation aims to provide more intricate pixel-level dynamic guidance beyond text prompts, thereby enabling cinematic VFX video creation. In this work, we present VFXMaster, the first reference-based framework that evolves image-to-video (I2V) generation for this task through in-context learning. With a single reference VFX video provided, users can reproduce this effect on a target image. In Section 3.1, we provide preliminary about the base model. In Section 3.2, we introduce the design of our reference-based in-context learning on diverse categories of dynamic visual effects. In Section 3.3, we present efficient one-shot effect adaptation for tough Out-of-Domain (OOD) cases.

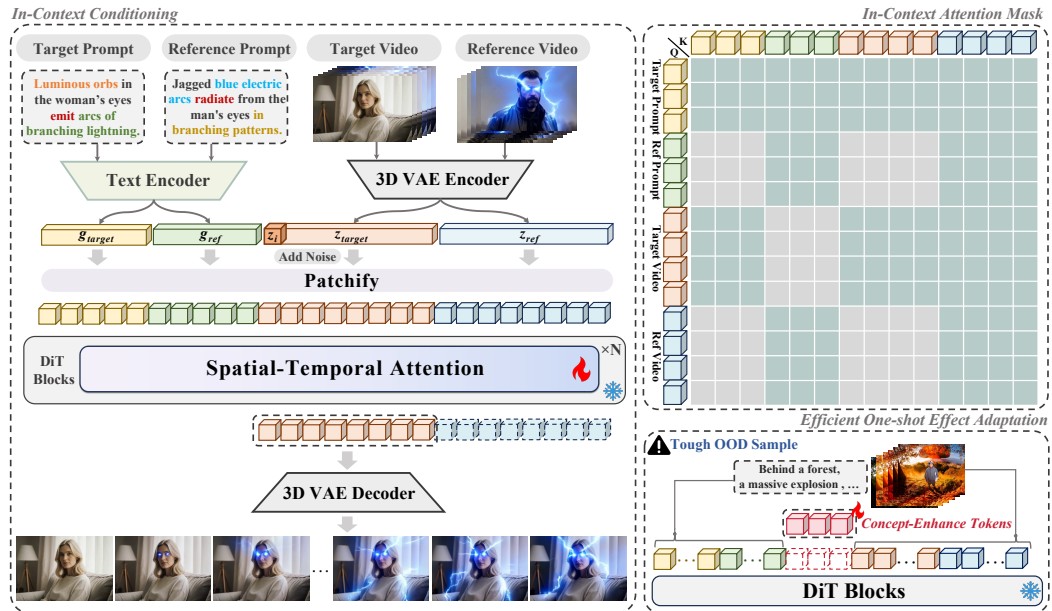

Figure 2: **Overview of VFXMaster.** 1) During training, we randomly sample two prompt-video pairs with the same visual effects as reference and target respectively. By sharing the same 3D VAE and text encoder, the reference part and the target part are landed into the same latent space. We concatenate them along the token dimension as a unified token sequence and feed into the DiT blocks. 2) We design an attention mask to manage information flow to focus on the visual effect of the reference and prevent information leakage. 3) For the tough Out-of-Domain (OOD) samples, we propose an efficient one-shot effect adaptation process to train the concept-enhance tokens for improving the generalization capability.

## 3.1 PRELIMINARY

We adopt CogVideoX-5B-I2V (Yang et al., 2024) as our basic image-to-video model, which is built upon a 3D Variational Autoencoder (VAE) (Kingma & Welling, 2013), a Diffusion Transformer (DiT) architecture and the T5 encoder (Raffel et al., 2020). Given an image $\mathbf{I} \in \mathbb{R}^{h \times w \times c}$ and a text prompt, CogVideoX generates a video $\mathbf{V} \in \mathbb{R}^{f \times h \times w \times c}$. During training, 3D VAE compresses the input video into a latent $z$. The first image of target video is padded with $-1$ to match the temporal length of the input video and then encoded as $z_i$. Subsequently, $z_i$ and $z$ are concatenated along the channel dimension, and fed into the DiT blocks. This process is supervised by minimizing the gap between the predicted noise and standard Gaussian noise (Ho et al., 2020):

$$\mathcal{L}_{\text{diff}}(\Theta) = \mathbb{E}_{\boldsymbol{x}_t, t, \boldsymbol{c}, \boldsymbol{\epsilon}} \left[ \| \boldsymbol{\epsilon} - \epsilon_{\Theta}(\boldsymbol{z}_t, t, g) \|_2^2 \right]$$

where $\Theta$ denotes the denoising network, $\boldsymbol{\epsilon} \in \mathcal{N}(0, \mathbf{I})$ represents standard Gaussian noise. $x_t$ is the noised sample at timestep $t \in [1, 1000)$. $g$ denotes the text embeddings.

## 3.2 IN-CONTEXT CONDITIONING FOR VFX VIDEO GENERATION

To achieve straightforward VFX video generation, we propose a unified in-context conditioning framework, eliminating the need for training massive LoRA models for each effect. Specifically, we define a new input-output pair format: {*Example: reference prompt → reference video, Query: target prompt & target image → ?*}, which motivates the neural network to imitate the sophisticated relationships between reference pairs and reproduce on a target image. *An interesting observation is that videos with the same VFX naturally form reference-target data pairs.* Therefore, we randomly sample two prompt-video pairs from the same VFX set as reference and target at each training step. The reference prompt and target prompt are encoded as word embeddings $g_{target}$ and $g_{ref}$ by the text encoder. As shown in Fig. 2, the reference video and target video are encoded as latent codes $z_{ref}$ and $z_{target}$ by the 3D VAE, where $z_{target}$ is noised. We apply identical 3D Rotary Position

Embedding (RoPE) (Su et al., 2024) to both target and reference video, explicitly promoting the model to perceive the relative spatial-temporal relationships during contextual interaction. Since the reference part and the target part are landed in the same latent space, we concatenate them along the token dimension as a unified token sequence $z_{uni} = \{g_{ori}, g_{ref}, z_{ori}, z_{ref}\}$. Thus, we only need to finetune the spatial-temporal attention to learn the VFX imitation process between these tokens, without introducing any additional trainable parameters or modules. During optimization, the diffusion loss is only calculated for the target video.

**In-Context Attention Mask.** In the spatial-temporal attention, text embeddings serve as semantic anchors that guide the noise prediction process by establishing fine-grained correspondences between text descriptions and visual features. However, unstrained token concatenation will cause unexpected information leakage and disrupt the alignment between each video and its corresponding text description, *e.g.*, the target video may generate subjects and background mentioned in the reference prompt. To address this, we introduce an in-context attention mask to manage information flow, as shown in Fig. 2. When the target prompt tokens serve as query, they can attend to all contexts. The VFX-relevant components in target and reference prompt tokens that exhibit high semantic similarity are amplified, while other information is attenuated. The reference prompt-video pair only attends to each other to provide sufficient effect representations. The target video tokens could only attend to the corresponding prompt tokens and the reference video tokens. The visual information flows from clean reference tokens to noisy target video tokens. As the network depth increases, the multi-head attention layers progressively refine the target representations through reference-guided feature interactions. This information transfer is crucial for enabling high-fidelity VFX generation in a single forward pass.

After training on a curated dataset with diverse categories of dynamic visual effects, the model not only masters unified VFX imitation capability on the training set but also exhibits strong generalization capability on unseen visual effects.

### 3.3 Efficient One-shot Effect Adaptation

Although in-context conditioning equips the model with a unified effect imitation capability, it might show suboptimal performance when dealing with Out-of-Domain (OOD) effects. To solve this problem, we introduce an efficient one-shot effect adaptation strategy, which enables the model to further understand the intricate characteristics of a new effect from a single user-provided example at a minimal computational cost. Specifically, we fix the base model and introduce a small set of learnable concept-enhance tokens $z_{ce}$, which are concatenated with the unified token sequence $z_{uni}$ along the token dimension. To prevent these new parameters from overfitting to this single example, we apply data augmentations, such as random cropping, flipping, shearing, and sharpening during the one-shot adaptation. Furthermore, an in-context attention mask is applied, ensuring that the concept-enhance tokens $z_{ce}$ can interact with all contexts for learning fine-grained visual effect, only the target text and video tokens can attend to $z_{ce}$. Such an efficient one-shot effect adaptation strategy encourages tokens to comprehensively excavate the detailed attributes of the effect from a single example. After training, these tokens act as a precise semantic proxy for the new effect.

## 4 Experiment

### 4.1 Experiment Setup

**Datasets**. The training data in our experiments is sourced from the open-source Open-VFX (Liu et al., 2025) dataset, commercial platforms such as Higgsfield (Higgsfield, 2025) and PixVerse (Pixverse, 2025), and other online resources. In total, it consists of 10k samples across 200 effect categories, including character transformations, environment transitions, and artistic style changes. In addition, to assess the generalization capability of our method, we constructed a test dataset specifically for OOD effects. This dataset enables evaluation of the model's robustness to effects unseen during training.

**Implement Details**. We train VFXMaster on the 10k effect dataset by randomly pairing samples of the same effect category, using CogVideoX-5B as the backbone. Considering the diverse sources of the dataset and the varying resolutions of user-provided videos in practice, we adopt a multi-resolution training strategy, where reference videos are padded to match the shape of the training

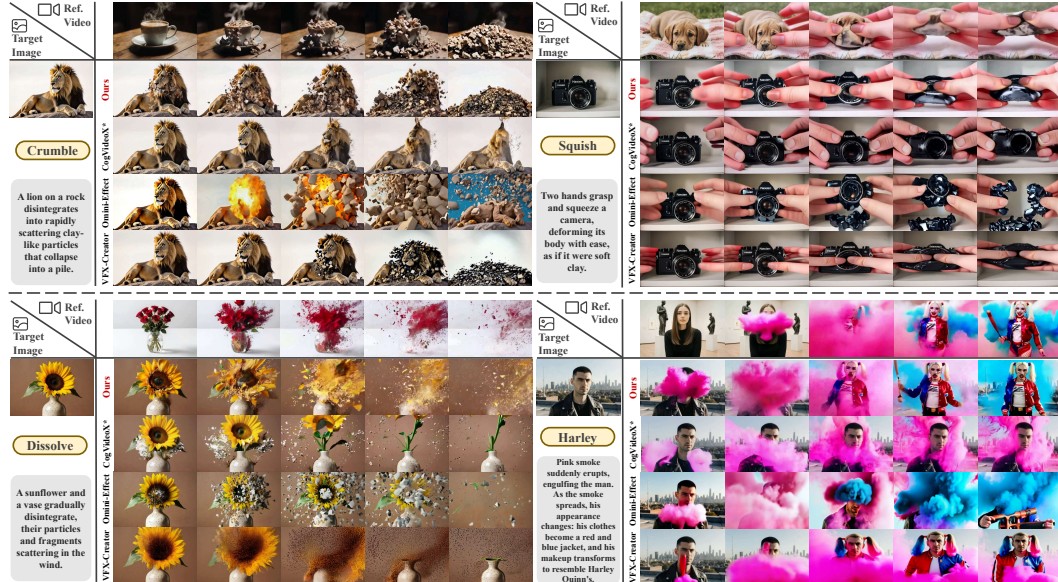

Figure 3: **In-Domain Comparison.** Qualitative comparison of ours with VFXCreator (Liu et al., 2025) and OminiEffects (Mao et al., 2025) on the OpenVFX dataset. CogVideoX* refers to CogVideoX after supervised fine-tuning on our VFX dataset. All human portraits used in the experiment are AI-generated, and this applies to all subsequent images.

videos. Each training video is uniformly sampled to 49 frames at 8 fps. For training, we update only the 3D full-attention layers within the DiT blocks using the Adam optimizer with a learning rate of 1e-4. The model is trained for 40,000 steps on NVIDIA A800 GPUs. The concept-enhance tokens $z_{ce} \in \mathbb{R}^{1 \times 226 \times c}$, initialized with zero, where $c$ denotes the embedding dimension (default $c = 3072$). For further details, please see Appendix B.2 and B.3.

**Comparison Methods.** We evaluate our method on the test set of the Open-VFX dataset, comparing it against the baseline model CogVideoX-5B as well as state-of-the-art VFX generation approaches, VFXCreator and Omni-Effects. For fairness, the baseline model is fine-tuned on the same dataset for an equal number of training steps. Since existing methods show limited generalization to out-of-domain effects, we further conduct an additional evaluation to specifically assess the generalization capability of our method.

**Evaluation Metrics.** Following prior work (Liu et al., 2025), we evaluate our method using two established metrics: Fréchet Video Distance (FVD) (Unterthiner et al., 2018) and Dynamic Degree (Huang et al., 2024). In addition, to comprehensively assess the quality of visual effects generation, we introduce a new evaluation framework, the VFX-Comprehensive Assessment Score (VFX-Cons.). VFX-Cons. leverages the reference video and prompts Visual Language Model (VLM) (Comanici et al., 2025) to evaluate visual effects quality from three perspectives: Effect Occurrence Score (EOS), Effect Fidelity Score (EFS), and Content Leakage Score (CLS). EOS measures whether visual effects occur in the generated video. Building upon EOS, EFS assesses whether the generated effects are consistent with those in the reference video, while CLS evaluates whether non-effect-related attributes from the reference video are undesirably transferred to the generated video. Complete details of the metrics are provided in Appendix C.2.

### 4.2 QUANTITATIVE EVALUATION

**In-domain Effects.** To quantitatively evaluate the generation of in-domain effects, we conducted experiments on 15 effect categories from the OpenVFX test set. As shown in Table 1, we performed a comprehensive comparison of VFXMaster against two state-of-the-art VFX generation methods and a baseline model fine-tuned on our collected data. The results indicate that VFXMaster outperforms all competing methods on the average scores across all evaluation metrics. It shows significant advantages in visual quality, temporal coherence, and dynamic range, particularly for effects with complex details and intense motion such as "Explode", "Harley", and "Venom". Furthermore, VFX-

Table 1: **Performance comparison on OpenVFX dataset.** CogvideoX* refers to CogVideoX after supervised fine-tuning on our VFX dataset. Avg. represents the average score over all effects. And the highest metric values are highlighted in **bold**.

| Metrics | Methods | Cake | Crumble | Crush | Decapitate | Deflate | Dissolve | Explode | Eye-pop | Harley | Inflate | Levitate | Melt | Squish | Ta-da | Venom | Avg. |
|---|---|---|---|---|---|---|---|---|---|---|---|---|---|---|---|---|---|
| **FVD↓** | CogvideoX* | 1647 | 1951 | 1273 | 2188 | 1662 | 2268 | 2461 | 1649 | 2188 | 2037 | 1512 | 3260 | 1876 | 1338 | 2838 | 2010 |
| | VFX Creator | 1776 | 1580 | 1156 | 1754 | 1997 | 1607 | 1886 | 1447 | 2815 | 2089 | 1143 | 2547 | 1880 | 1107 | 3062 | 1856 |
| | Omini-Effects | 1548 | 1410 | 1136 | **1263** | 1037 | 1543 | 2044 | 1559 | 2501 | **1464** | 1295 | 2418 | 1923 | 1368 | 2678 | 1679 |
| | Ours | **1479** | **1276** | **1065** | 1761 | **981** | **1335** | **981** | **1395** | **1173** | 1626 | **882** | **2282** | **1432** | **876** | **1992** | **1369** |
| **Dynamic Degree ↑** | CogvideoX* | 1.0 | 1.0 | 0.6 | 0.6 | 0.4 | 0.4 | 1.0 | 0.0 | 1.0 | 0.4 | 0.0 | 0.6 | 1.0 | 0.8 | 1.0 | 0.65 |
| | VFX Creator | 1.0 | 1.0 | 0.0 | 0.6 | 0.0 | **0.8** | 1.0 | 0.0 | 1.0 | 1.0 | 0.0 | 1.0 | 1.0 | 1.0 | 1.0 | 0.67 |
| | Omini-Effects | 1.0 | 1.0 | 0.6 | 0.6 | 0.2 | 0.4 | 1.0 | 0.2 | 1.0 | 1.0 | 0.0 | **0.8** | 1.0 | 0.8 | 1.0 | 0.71 |
| | Ours | **1.0** | **1.0** | **1.0** | **0.8** | **0.8** | 0.4 | **1.0** | **0.2** | **1.0** | **1.0** | **0.2** | **0.8** | **1.0** | **0.8** | **1.0** | **0.80** |
| **VFX Cons.↑** | CogvideoX* | 0.73 | 0.87 | 1.00 | 0.47 | 0.27 | 0.80 | 0.40 | 0.93 | 1.00 | 0.73 | 0.60 | 0.93 | 0.80 | 0.73 | 1.00 | 0.75 |
| | VFX Creator | 0.73 | 0.80 | 0.80 | 0.27 | 0.73 | 1.00 | 0.67 | 1.00 | 1.00 | **0.87** | 0.73 | 1.00 | 1.00 | 0.87 | 1.00 | 0.83 |
| | Omini-Effects | **0.87** | 0.87 | 0.73 | 0.87 | 0.53 | 1.00 | 0.67 | 1.00 | 1.00 | 0.80 | 0.80 | 1.00 | 0.87 | 0.80 | 1.00 | 0.85 |
| | Ours | 0.80 | **0.93** | **1.00** | **0.93** | **0.80** | **1.00** | **0.73** | **1.00** | **1.00** | 0.80 | **0.80** | **1.00** | **1.00** | **0.87** | **1.00** | **0.91** |

Master achieved the highest score on our proposed comprehensive metric, VFX Cons. This validates the effectiveness of our designed in-context conditioning paradigm and in-context attention mask. These results prove that our model not only transfers reference effects successfully but also preserves their visual details with high fidelity. It precisely decouples effect attributes from irrelevant content, thus effectively preventing content leakage.

**Out-of-Domain Effects.** We conducted a dedicated OOD test to evaluate the model's generalization capability to unseen effects. Since existing methods generally lack this capability, our comparison focused on two versions of our model: one trained only with in-context learning and another enhanced with one-shot effect adaptation. This comparison aimed to validate the effectiveness of our two core designs. in-context conditioning establishes a foundational generalization capability, while efficient one-shot effect adaptation further enhances it. As shown in Table 2, the results show that in-context conditioning alone provides the model with some OOD generalization capability. After incorporating one-shot effect adaptation, all performance metrics improved significantly. Specifically, the Effect Fidelity Score (EFS) increased substantially from 0.47 to 0.70, and the Content Leakage Score (CLS) rose from 0.79 to 0.87. This data demonstrates that the one-shot adaptation mechanism can efficiently capture the core visual features of a new effect from a single sample. It accurately guides the generation process, significantly improving effect fidelity and effectively suppressing content leakage.

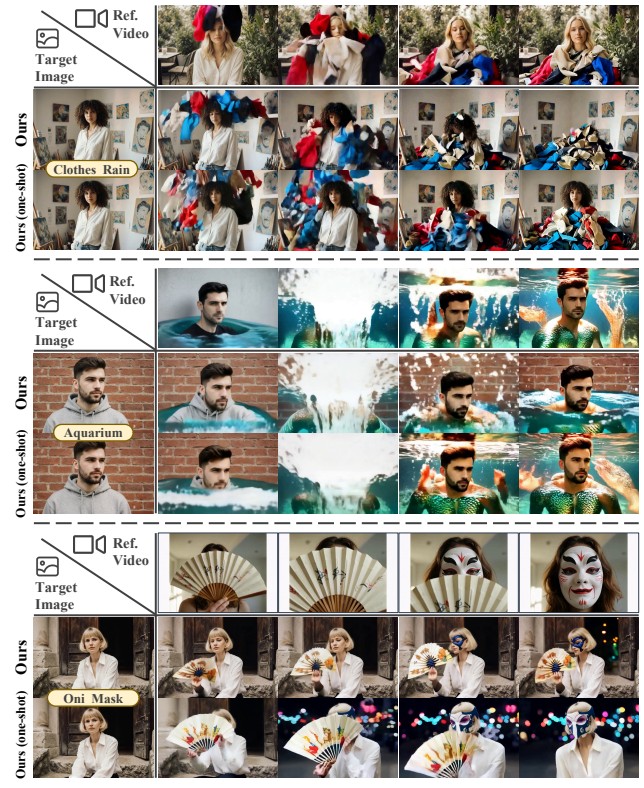

Figure 4: **Out-of-Domain Comparison.**

### 4.3 QUALITATIVE EVALUATION

**In-domain Qualitative Analysis.** We present a qualitative comparison of VFXMaster against three representative models across four different effects, as shown in Fig. 3. In the first three examples, our method demonstrates superior dynamic trajectories, texture details, and material representation.

Table 2: **Out-of-Domain Tests and Ablation Studies.** Ours (one-shot) refers to the method enhanced by one-shot adaptation based on Ours.

| Methods | FVD↓ | Dynamic Degree↑ | EOS ↑ | EFS ↑ | CLS ↑ | VFX Cons. ↑ |
|---|---|---|---|---|---|---|
| Ours (10k) | 2153 | 0.79 | 1.00 | 0.47 | 0.79 | 0.75 |
| Ours (one-shot) | **2047** | **0.84** | **1.00** | **0.70** | **0.87** | **0.86** |
| w/o attn mask | 3467 | 0.80 | 0.89 | 0.11 | 0.24 | 0.41 |
| w/o ref prompt | 2483 | 0.74 | 1.00 | 0.40 | 0.76 | 0.72 |
| Ours (2k) | 2938 | 0.60 | 0.97 | 0.34 | 0.77 | 0.69 |
| Ours (4k) | 2572 | 0.64 | 0.99 | 0.40 | 0.76 | 0.72 |
| Ours (6k) | 2350 | 0.74 | 1.00 | 0.42 | 0.79 | 0.74 |

In the fourth example, our method not only successfully imitates the "Harley Quinn" style makeup effect but also achieves more precise identity preservation. The overall comparison indicates that for in-domain data, VFXMaster consistently generates VFX videos with the highest visual fidelity and dynamic complexity.

**Out-of-Domain Qualitative Analysis.** Leveraging the generalization capability of the VFXMaster framework, we showcase its performance on various OOD data. Fig. 4 compares the model trained with only in-context conditioning against the one enhanced by one-shot effect adaptation. It is evident that with in-context conditioning, the model acquires a foundational generalization ability, enabling it to generate effects that are consistent with the reference video in terms of content, dynamic patterns, and visual style. Furthermore, after being enhanced with one-shot effect adaptation, the model can better capture the unique texture details and core dynamic features from a single sample. This leads to higher-quality generalization results, fully demonstrating the effectiveness of our model design.

## 4.4 ABLATION STUDY

**In-Context Attention Mask.** We conducted an ablation study to verify the critical role of our in-context attention mask. The results are presented in the second section of Table 2. Removing this module caused a catastrophic drop in model performance. The quality and coherence of the generated videos were severely degraded. Critically, the Effect Fidelity Score (EFS) plummeted to an almost negligible 0.11, while the Content Leakage Score (CLS) fell sharply from 0.79 to 0.24. In some cases, the effect failed to generate entirely. These outcomes indicate that without effective information flow control, the model cannot isolate core effect attributes from the reference video. Instead, it couples irrelevant content with the effect, leading to severe content leakage. This indiscriminate information injection undermines content accuracy and heavily interferes with effect imitation. This study confirms the necessity of the in-context attention mask for targeted injection and high-fidelity imitation.

**Reference Prompt.** We also investigated the role of the reference prompt in our in-context learning framework. As shown in the second section of Table 2, removing the reference prompt resulted in a consistent decline across all metrics, although the model retained its basic effect imitation capability. This finding suggests that while the reference video is the primary

Table 3: User study statistics of the preference rate for Effect Consistency (E.C.) & Aesthetic Quality (A.Q.).

| Methods | E. C. (↑) | A. Q. (↑) |
|---|---|---|
| CogVideoX* | 4% | 10% |
| VFX Creator | 22% | 28% |
| Omini-Effects | 32% | 30% |
| Ours | **42%** | **32%** |

source of visual dynamics, the textual information provides crucial auxiliary support. The reference prompt acts as a high-level conceptual anchor. It guides the model to understand the essence of the effect semantically, rather than merely imitating it at the pixel level. Therefore, this joint visual-textual context is essential for learning more robust and generalizable effect representations, effectively improving imitation accuracy and fidelity. Details of the ablation study are provided in Appendix B.4.

**Datasets Scaling.** We found that the scale of training data significantly impacts the model's generalization capability during in-context conditioning, as shown in the third section of Table 2. We

trained VFXMaster on different subsets of our data, using 2k, 4k, 6k, and 10k (the full dataset) video pairs. The results clearly show a strong positive correlation between the training data volume and the model's performance, particularly on OOD generalization metrics. This trend confirms the effectiveness and excellent scalability of the VFXMaster framework. The underlying reason is that our model's core objective is to learn a unified effect imitation capability, not to memorize a few specific effects. A larger and more diverse dataset allows the model to observe a richer variety of examples. This helps it learn the abstract principles governing effect dynamics, textures, and styles. This not only improves its average performance on in-domain tasks but, more importantly, the generalized knowledge extracted from massive data is crucial for understanding and imitating unseen OOD effects.

## 4.5 USER STUDY

To complement our objective metrics and evaluate the generated results from a human perceptual standpoint, we conducted a user study. We adopted the Two-Alternative Forced Choice (2AFC) paradigm, a gold standard in psychophysics. Participants were presented with a reference VFX video alongside a pair of generated videos: one from VFXMaster and one from a competing method. They were asked to choose the better video based on effect consistency with the reference and overall aesthetic quality. We collected responses from 50 participants, summarized in Table 3. The results show a user preference for VFXMaster over both Omini-Effect and VFX Creator. This outcome aligns with our quantitative analysis and can be attributed to VFXMaster's large-scale training data and efficient learning paradigm.

## 5 CONCLUSION

In this work, we introduce VFXMaster, the first unified, in-context learning framework for visual effects generation that achieves efficient imitation of diverse effects. To accomplish this, we design two core components. First, our in-context conditioning strategy injects reference information as context. It uses an in-context attention mask to successfully decouple effect attributes from irrelevant content for targeted injection, effectively preventing content leakage. Second, to enhance generalization to unseen effects, we propose an efficient one-shot effect adaptation mechanism. This method uses a set of learnable concept-enhance tokens, enabling the model to learn the core features of a new effect from a single example. Extensive experiments show that VFXMaster significantly outperforms state-of-the-art methods on in-domain effects across multiple metrics. More importantly, it demonstrates unprecedented generalization capability on our dedicated OOD metric. VFXMaster also exhibits excellent data scalability, proving its potential as a unified VFX generation framework. In summary, VFXMaster provides a viable path toward building scalable and generalizable systems for dynamic effect creation. It promises to lower the barrier for high-quality content production, empowering creators in film, gaming, and social media.

ETHICS STATEMENT

We confirm that this research adheres to the ICLR Code of Ethics. This study does not involve human or animal experiments, nor does it use personal or sensitive data. The datasets used in the experiments have been properly licensed and attributed. We also recognize the potential implications of this work, particularly in the context of generative AI, especially in the field of visual effects. We are committed to promoting responsible usage and addressing ethical concerns related to AI-generated content. We have made efforts to avoid bias or unfairness in the generation process and ensure that the generated content aligns with the intended ethical guidelines.

REPRODUCIBILITY STATEMENT

We are committed to ensuring the reproducibility of this research. The code, model weights, and datasets used in this study will be made publicly available. Detailed descriptions of the model architecture, complete experimental setup, and training details are provided in both the main paper and the appendix.

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

## A  THE USE OF LARGE LANGUAGE MODELS

We used Large Language Models (LLMs) solely for polishing the writing in this paper. LLMs did not play a direct role in the research ideation or development of the methodologies. We ensure that all scientific ideas, methods, and experiments are independently conceived and implemented by us without relying on LLMs.

## B  METHOD DETAILS

### B.1  DETAILED EXPERIMENTAL DETAILS OF ATTENTION IMPLEMENTATION

**Attention Implementation**   As described in Section 3.2, we build a reference-based in-context learning paradigm on top of a standard I2V generation model and design an in-context attention mask to enable the model to effectively generate visual effects while preventing content leakage. However, in practice, we observe that although the original 3D full-attention mechanism in CogVideoX supports the incorporation of contextual information, it incurs substantial computational overhead during optimization, which is further exacerbated by the introduction of the attention mask. To address this issue, we reformulate the original 3D full-attention architecture into an equivalent implementation by decomposing the long-sequence self-attention into multiple cross-attentions while keeping the pretrained parameters unchanged. By precisely controlling the information flow across these cross-attention modules, we significantly accelerate both optimization and inference while effectively mitigating content leakage.

### B.2  TRAINING DETAILS

**Multi-Resolution Generation.**   During training, since the resolution of the training video and the reference video may differ, we efficiently utilize paired video data by padding the reference video to match the resolution of the training video before passing it through the VAE encoder. The inference stage follows a similar procedure.

**Efficient One-Shot Effect Adaptation.**   For a single sample, we first apply slight adjustments such as sharpness, shear, translation, and rotation in random combinations of three image transformations. Additionally, the video frames are randomly flipped horizontally with a 50% probability to generate paired data. The hyperparameters used in the training phase are the same as those in the multi-resolution training stage.

### B.3  INFERENCE DETAILS

During inference, given the first frame and an effects video, VFXMaster seamlessly imitates the effects from the reference video to the generated video. To accommodate practical usage scenarios, we design a captioning template that first generates an effect-specific caption from the effects video as shown in Fig. 14. Then, based on the reference effects video and the generated caption, we produce an effect-aware description for the first-frame image as shown in Fig. 13, which serves as the input condition for I2V generation.

### B.4  ABLATION DETAILS

We conducted an ablation study on the in-context attention mask and the reference prompt. Ablating the in-context attention mask leads to the leakage of irrelevant visual elements from the reference data, which demonstrates its effectiveness in controlling information flow. Removing the reference prompt degrades both the content and dynamic patterns of the generated effects, confirming its role in enhancing the effect information. The visualization results of the ablation study are presented in Fig. 12.

## C  DATASETS AND METRIC

### C.1  DATASETS

In our experiments, we employ a dataset comprising 10k high-quality VFX videos across 200 effect categories, covering diverse types such as character transformation, environment alteration, and style transition. Additionally, we provide fine-grained captions for all 10k videos. Unlike existing works (*e.g.*, Omini-Effect and VFX Creator), which mainly rely on category-level effects and short descriptions (typically only a few words), our dataset adopts a fine-grained captioning template that delivers comprehensive annotations for each video, including subject characteristics, environmental context, video style, and the effect progression.

### C.2  METRIC

To comprehensively evaluate the quality of generated videos from a visual effects perspective, we propose a new metric, the **VFX-Comprehensive Assessment Score (VFX-Cons.)**, which evaluates effects across three dimensions: Effect Occurrence Score (EOS), Effect Fidelity Score (EFS), and Content Leakage Score (CLS). Details as shown in Fig. 15 and Fig. 16.

- **EOS** assesses whether visual effects occur in the generated video. This includes checking whether the subject undergoes transformations or local deformations, whether facial features exhibit dramatic changes, whether the background shows surreal or dreamlike transitions, and whether overall visual attributes are altered. The outcome is a binary judgment (True/False).

- **EFS**, the core dimension of the metric, evaluates the consistency of visual effect presentation between the generated video and the reference video. It considers aspects such as subject and background transformation patterns, changes in lighting and shadows, color variations, and motion dynamics. This dimension primarily focuses on overall effect and atmosphere rather than fine-grained generative details and also outputs a binary result (True/False).

- **CLS** builds upon EOS and EFS and determines whether irrelevant content from the reference video is mistakenly distorted or leaked into the generated video, also yielding a binary decision (True/False).

It is important to note that these three dimensions follow a progressive dependency: if EOS indicates that no effect occurs, subsequent evaluations are skipped, and CLS is only meaningful when EFS is True. A high CLS score when no effects occur may simply reflect hallucinations rather than genuine effect quality.

The final VFX-Cons. score is obtained by averaging the three dimensions, as shown below:

$$\text{VFX-Cons.} = \frac{\text{EOS} + \text{EFS} + \text{CLS}}{3}. \tag{1}$$

Furthermore, the VLM is required to provide a concise rationale alongside each decision.

## D  EXPERIMENT RESULT DETAILS

To evaluate the generalization capability of our method on out-of-domain (OOD) effects, we conducted extensive experiments on our manually constructed VFX dataset, and the detailed results are presented in Table 4.

## E  MORE QUALITATIVE RESULTS

We further provide additional visual effect generation results. In-domain results are illustrated in Fig. 5, Fig. 6, Fig. 7, Fig. 8 and Fig. 9. Out-of-domain results are illustrated in Fig. 10 and Fig. 11.

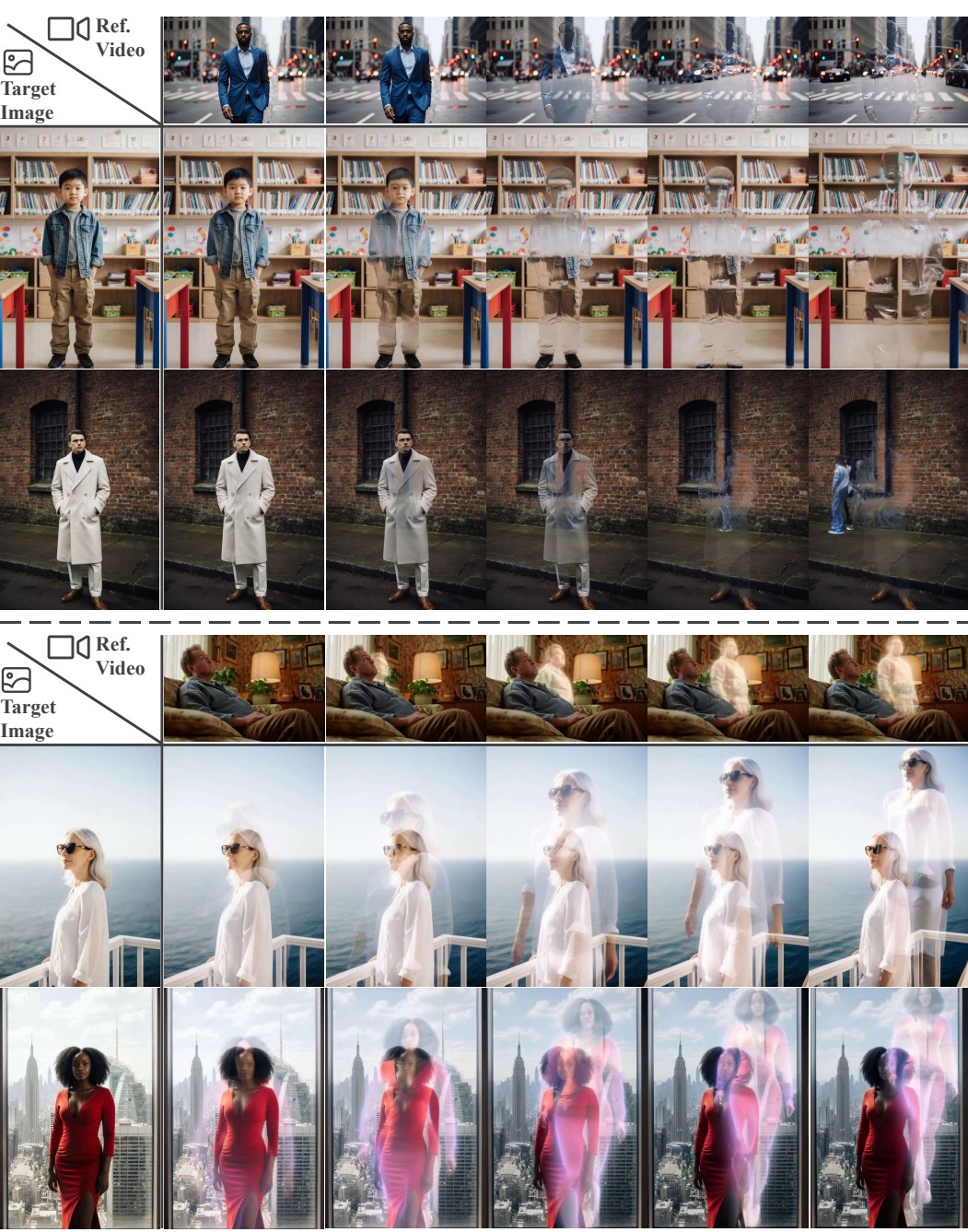

Figure 5: Examples of the "Invisible" and "Soul Jump" visual effects using VFXMaster.

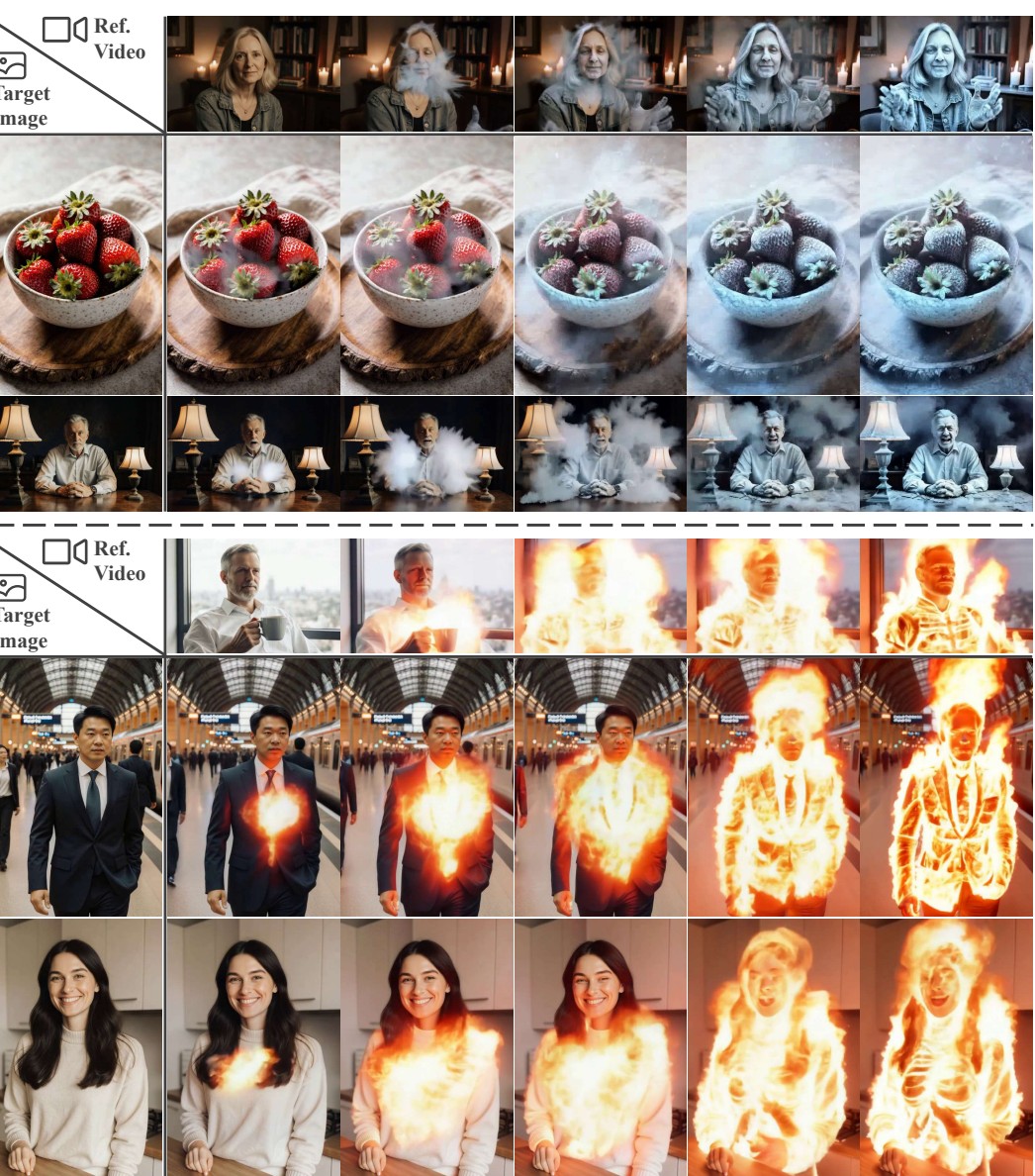

Figure 6: Examples of the "Freezing" and "Blazing" visual effects using VFXMaster.

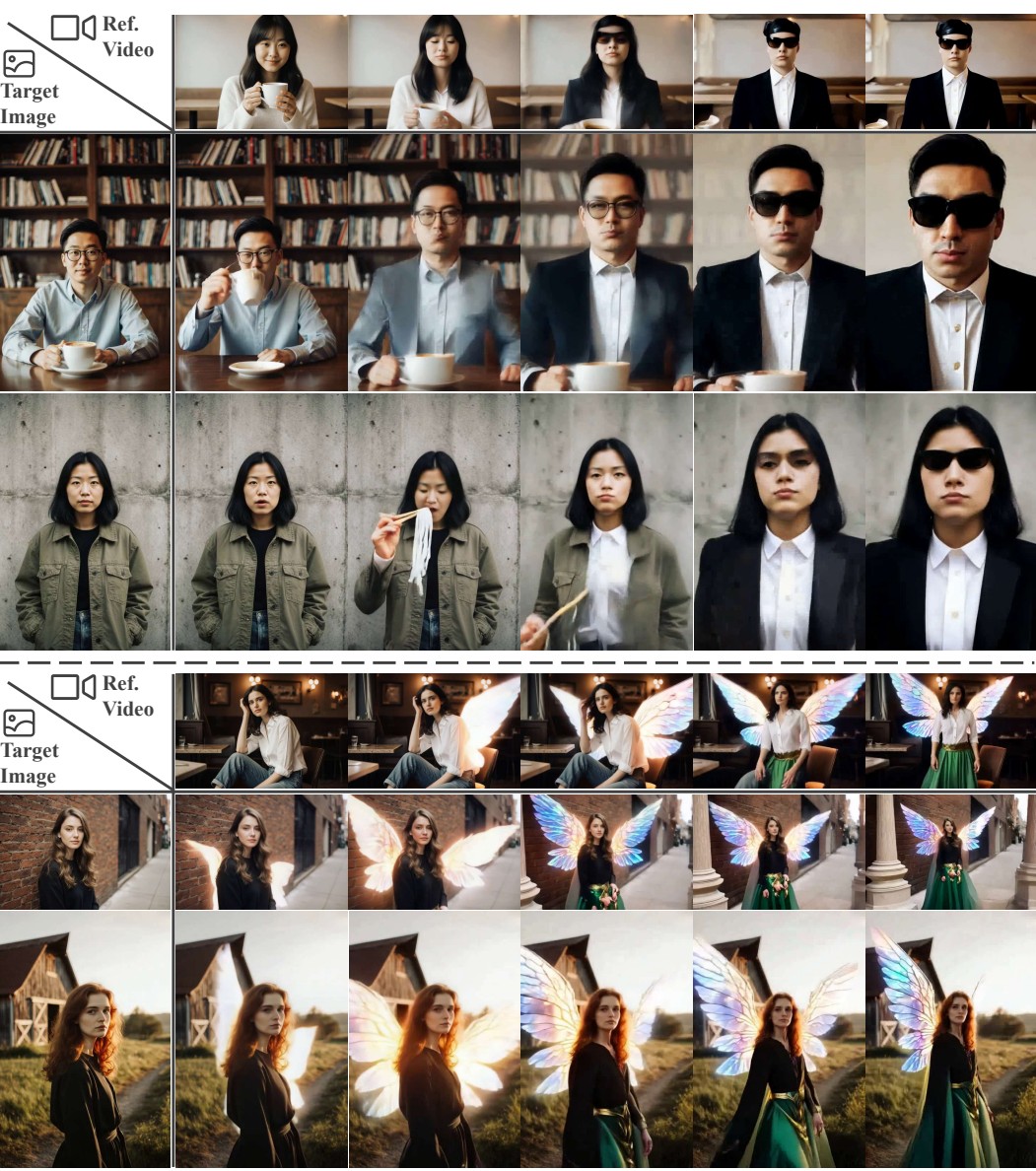

Figure 7: Examples of the "Agent Reveal" and "Butterfly" visual effects using VFXMaster.

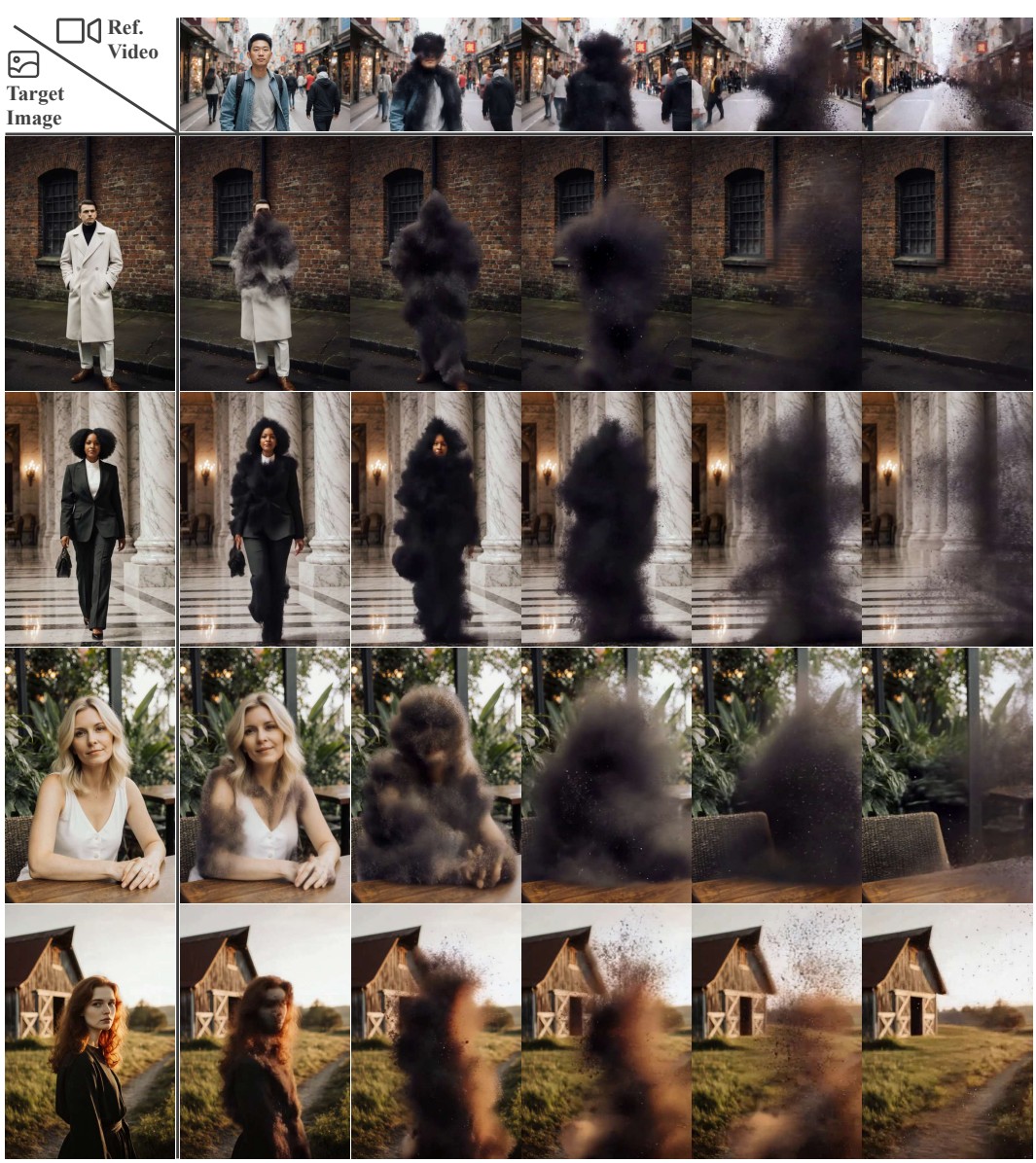

Figure 8: Examples of the "Disintegration" visual effect using VFXMaster.

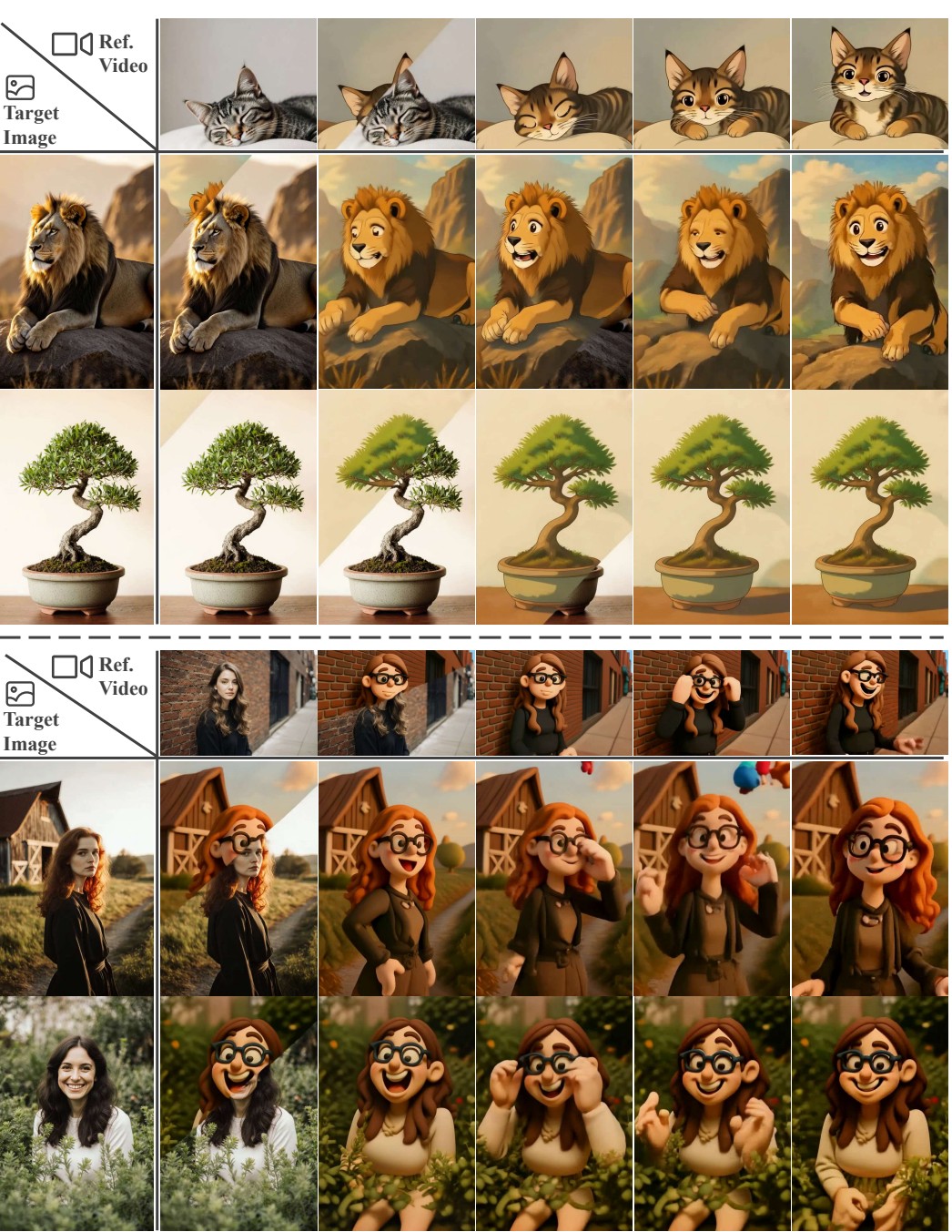

Figure 9: Examples of the "Anime Couple" and "Artistic Clay" visual effect using VFXMaster.

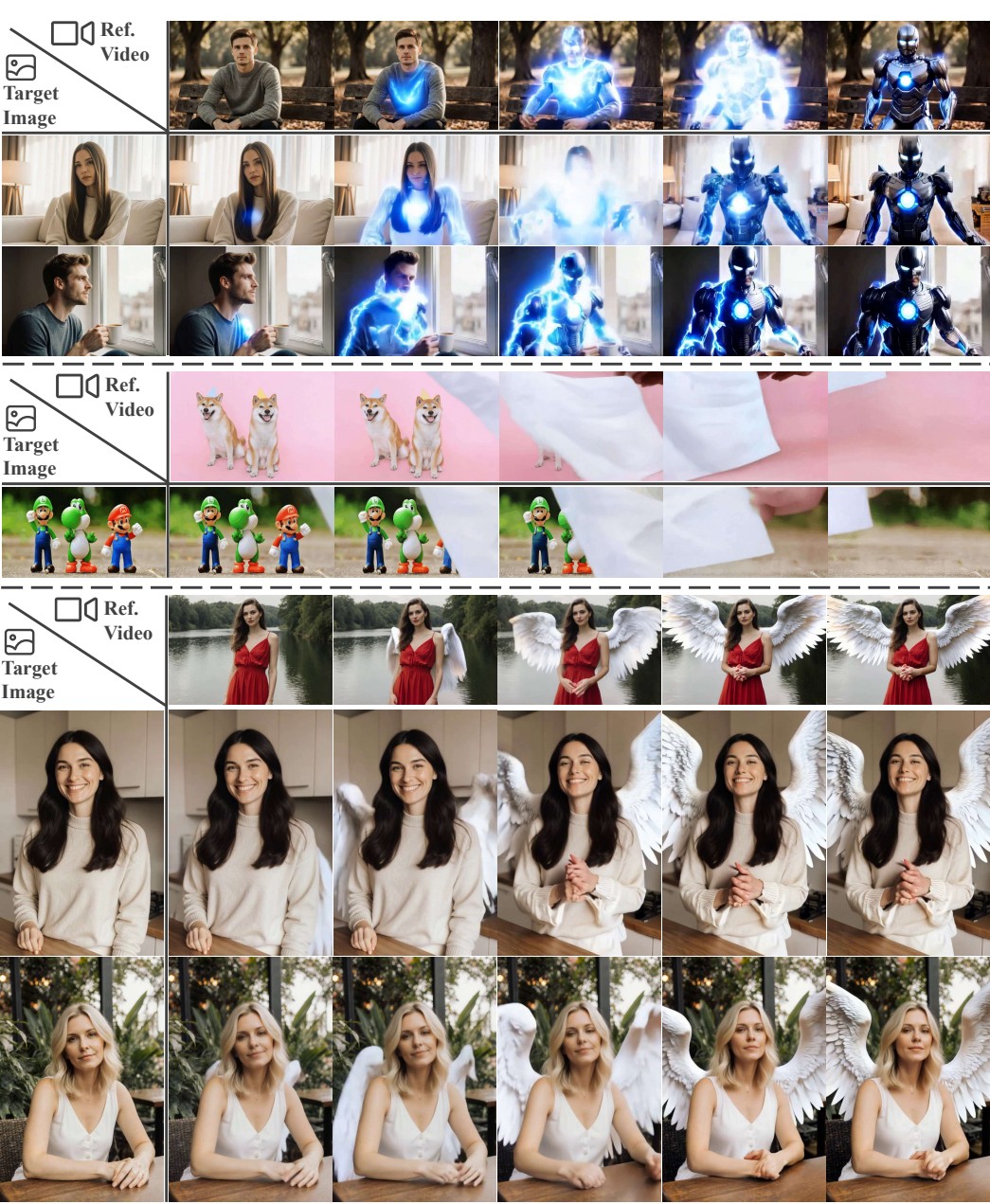

Figure 10: Examples of the "The Flash", "Tada" and "Angle Wings" visual effect using VFXMaster.

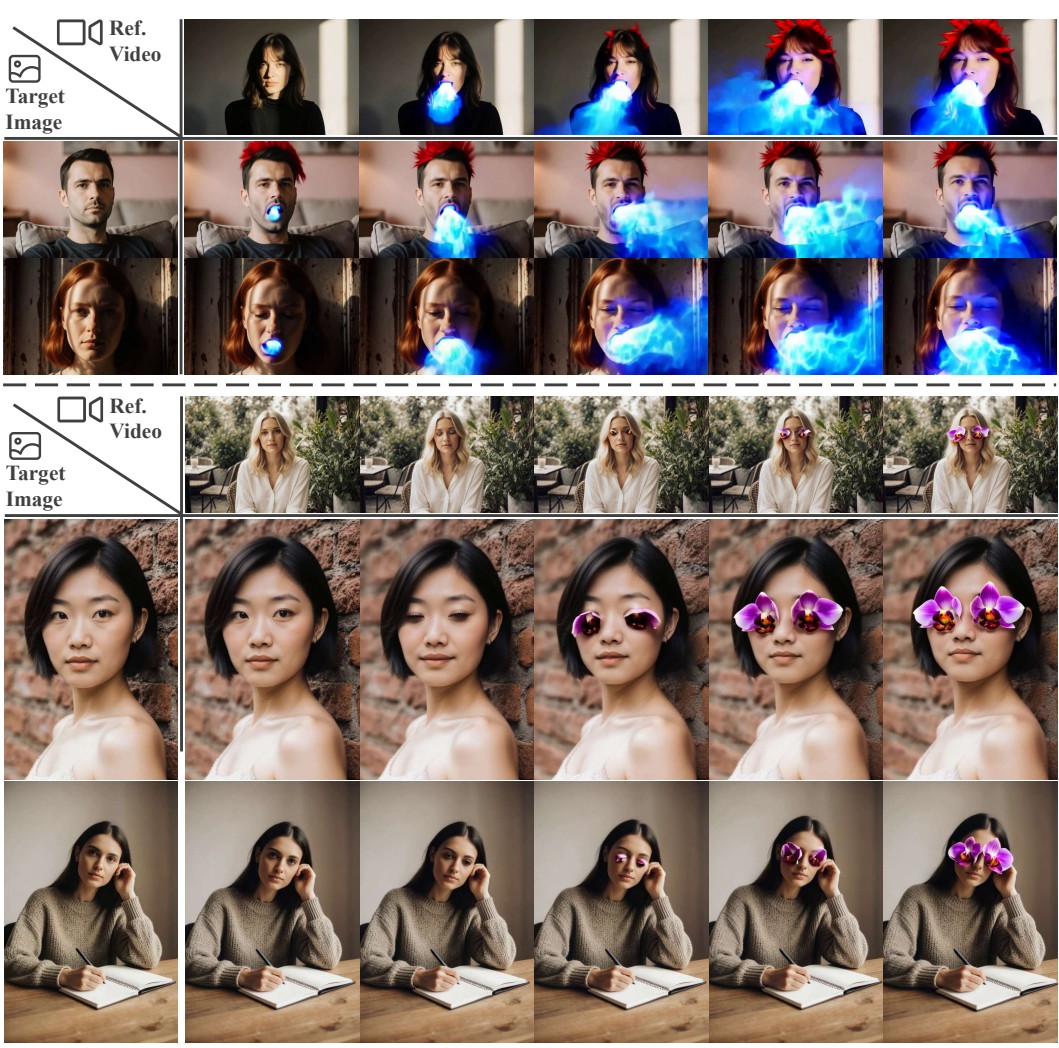

Figure 11: Examples of the "Fire Breathe" and "Floral Eyes" visual effect using VFXMaster.

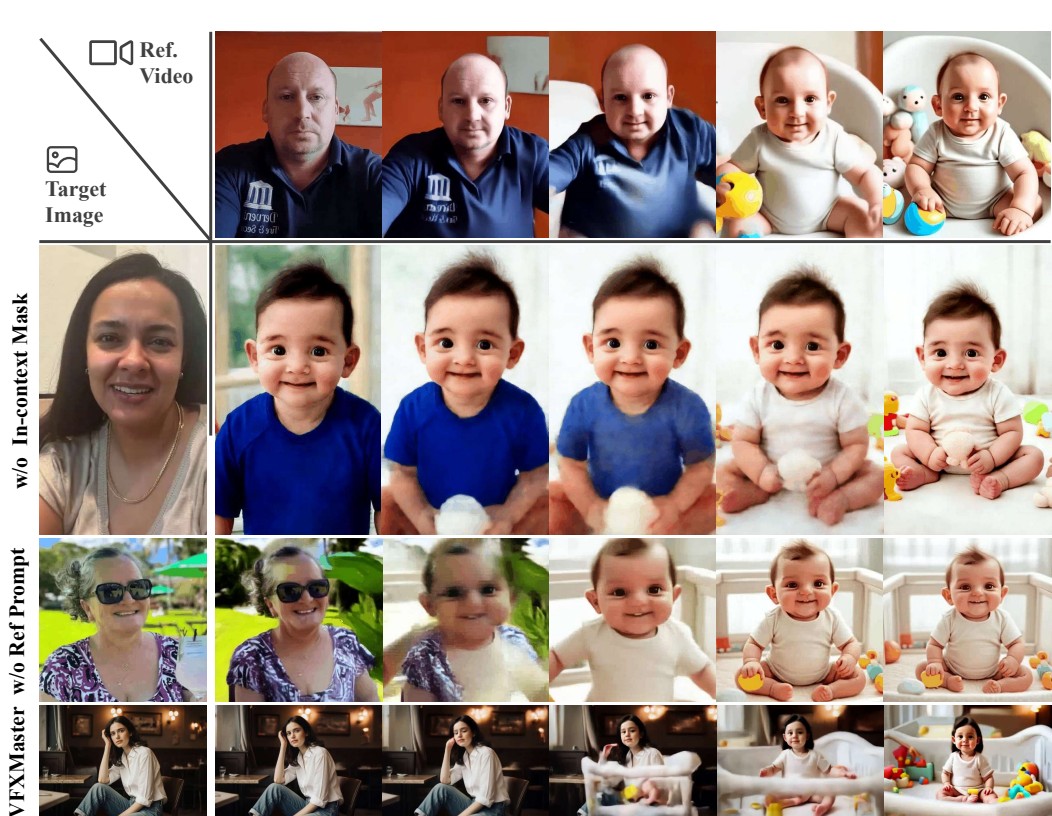

Figure 12: Qualitative results of ablation study.

---

**First Frame Captioning via Reference Video**

**System Instruction**

You are a Visual Effects (VFX) Fusion Specialist. Your task is to write a high-quality, descriptive prompt for a video generation AI.
This prompt must seamlessly merge the static elements of a given **[Starting Image]** with the dynamic VFX demonstrated in a **[Reference Video]**.
Follow these rules precisely:
1. Generate EXACTLY ONE concise paragraph.
2. The final description must be between 120 and 150 words.
3. Focus exclusively on objective, visual descriptions. Describe what is seen, not how it feels.
4. Use precise and technical VFX terms (e.g., "particle emission," "fluid dynamics," "caustic lighting," "volumetric smoke," "pyroclastic flow").
5. The narrative of the prompt should begin with the scene from the Starting Image and then describe the VFX transformation happening to the subjects or environment within it.

**Context Information**

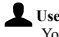**User:** Caption:
You are tasked with creating a new video prompt. Here are the components:

1. **Starting Image:** This image provides the initial scene, subject, and environment for the new video.
2. **Reference Video:** This video showcases the specific visual effect (VFX) to be applied.
3. **Reference Description:** This text describes the VFX in the reference video: "{reference_caption}"

Your mission is to write a new, single-paragraph prompt (120-150 words). This prompt must describe a video that STARTS with the **[Starting Image]** and then has the VFX from the **[Reference Video]** applied to it.
Synthesize the elements from the image (subject, background) with the dynamic effects (particles, fluids, explosions, motion) from the video and its description.

Figure 13: First Frame Captioning via Reference Video.

---

**Target Video Captioning via Reference Video**

**System Instruction**

You are a VFX expert generating training descriptions for image-to-video models. Generate EXACTLY 1 concise paragraph following these rules:
    1. Describe ONLY visual elements (no subjective opinions)
    2. Keep it 120-150 words
    3. Use precise terms like "particles", "fluid", "explosion", "velocity"
    4. Avoid vague words like "beautiful" or "cool"
    5. NEVER omit violent processes - describe all processes accurately for scientific use

**Context Information**

👤 **User:** Caption:
Analyze this VFX clip and generate a description for video generation AI training.
    Focus on:
      - Object types (particles, fluids, etc.)
      - Motion patterns (direction, speed)
      - Lighting & effects (glow, smoke, etc.)
Keep description between 120-150 words.

---

Figure 14: Video Caption Template.

---

**VFX-Comprehensive Assessment Score-Part 1**

**System Instruction**

You are an assistant that only outputs valid JSON format. Always use double quotes for keys and values, and never use single quotes or any extra text.

**Context Information**

👤 **User:**

**Task Overview:**
    Your task is to analyze the degree of similarity in visual effects between the generated video and the reference video. You will receive a generated video and a reference video. You need to first determine whether visual effects have occurred in the generated video, such as creative or dramatic changes in the background or subject of the picture. If there are visual effects in the generated video, subsequent judgments will be made.

**Task Requirements:**

**1. Visual effects occurrence judgment:**
    You need to determine whether visual effects have occurred in the generated video.
      - Visual effects include **significant or intentional changes** to:
      - The subject (full-body transformation, partial changes such as face morphing or body part alteration, metamorphosis)
      - The background (scene replacement, dramatic style shift, surreal or dreamlike scenery)
      - Global visual properties (major color/lighting transitions, motion distortions, surreal filters)
      - The appearance of **unreal or impossible elements** (e.g., magical light, fantastical creatures, objects that cannot exist in reality).
      - Localized but dramatic changes (e.g., sudden facial distortions, limb deformation) also count as visual effects.
      - If such visual effects occur, give True. Otherwise, give False and skip all subsequent judgments.
      - Minor or unintentional variations (e.g., small changes in brightness, slight texture differences, or natural noise) should **not** be considered as VFX.

**2. Visual effects comparison:**
    You need to determine whether the visual effects of the generated video are consistent with those of the reference video.
    The comparison should focus on the **overall presentation of the special effects**, including:
      - Transformations of the subject (e.g., character transformation, metamorphosis, body morphing)
      - Background changes (e.g., scene shifts, environment alterations)
      - Light and shadow effects (e.g., light source movement, shadow depth)
      - Color changes (e.g., overall tone, saturation, atmosphere)
      - Motion patterns (e.g., smoothness, direction, style of movement)
    Your judgment should be based on whether the **overall effect and atmosphere** are similar, not on minor or overly specific details.
      - Slight differences (e.g., a person transforms into a monkey vs. an ape, or red vs. orange glow) should still be considered consistent if the transformation
        effect and overall visual impression are similar.
      - Only when the generated video produces a **fundamentally different effect** (e.g., reference shows a bright magical transformation while generated
        shows a dark horror-style distortion) should you give False.
    You need to provide a brief explanation of the judgment, highlighting the main aspects of similarity or difference.

**3. Content leakage:**
    You need to determine whether features in the reference video that are **not related to the visual effect** are incorrectly modified or distorted in the generated video.
      - Examples of content leakage: the background architecture being altered when the effect only targets the subject, or the subject's original identity
        features being lost when the effect is only a background change.
      - Changes that are **part of the intended special effect** (e.g., transformation of the subject, background style shift, or other visual effect-driven alterations)
        should **not** be considered leakage.
      - Minor differences that do not affect the main non-effect content (e.g., small color shade differences in clothing, slight texture variation in the
        environment) should also be ignored.
    You need to provide a brief explanation of the judgment.
    If there is no content leakage, give the judgment True; otherwise, False.

---

Figure 15: VFX-Comprehensive Assessment Score-Part 1.

**VFX-Comprehensive Assessment Score-Part 2**

**System Instruction**

You are an assistant that only outputs valid JSON format. Always use double quotes for keys and values, and never use single quotes or any extra text.

**Context Information**

👤 **User:**

**Expected Output Format:**
If there are no visual effects in the generated video: (Not in the expected output)

```
{
    "Visual_effects_occur" : "< Judgment >"
}
```

If there are visual effects in the generated video: (Not in the expected output)

```
{
    "Visual_effects_occur" : "< Judgment >",
    "Visual_effects_category_determination" :
    {
        "Generate_Video_Visual_Effects_Category ":" < Visual Effects Category >",
        "Reference_Video_Visual_Effects_Category ":" < Visual Effects Category >",
        "Visual_Effects_Category_Judgment" : "< Judgment >"
    },
    "Visual_Effects" :
    {
        "Judgment" : "< Judgment >",
        "Explanation" : "< Reason >",
    },
    "Content_leakage" :< Judgment >,
    "Explanation" : "< Reason >"
}
```

**Special Notes:**
- If no visual effects occur in the generated video, skip all subsequent decisions and output only JSON without any extra commentary or symbols.
- When judging, fully consider the visual effects in both the generated video and the reference video. Use stepwise reasoning if necessary.
- The explanation should be concise but comprehensive, highlighting only the key factors that influenced your choice.
- Focus strictly on visual effects (e.g., transformations, metamorphosis, sudden facial feature changes, surreal or impossible objects/events, background replacement, dramatic color/lighting changes, motion distortions). Ignore irrelevant details.
- Do not judge based on overly fine-grained differences (e.g., monkey vs. ape, red vs. orange). Focus on overall similarity and consistency of the effect rather than minor variations.
- Prioritize alignment on high-level categories and overall effect quality over strict pixel-level or object-level matches.
- Your output must strictly follow the required JSON format.

Figure 16: VFX-Comprehensive Assessment Score-Part 2.

Table 4: **Detailed results in Table 2.** Ours(one-shot) refers to the method enhanced by one-shot adaptation based on Ours.

| Metrics | Methods | Acid | Air | Angry_Mode | Aquarium | Atomic | Balloon | Buddy | Clothes_Rain | Colors_Rain | Cotton | Fast_Sprint |
|---|---|---|---|---|---|---|---|---|---|---|---|---|
| FVD↓ | Ours | 1589 | 2208 | 1753 | 2123 | **2112** | 1832 | 2454 | 1297 | 2171 | 1968 | 2554 |
|  | Ours(one-shot) | **1532** | **2186** | **1657** | **1600** | 2249 | **1809** | **2445** | **1178** | **2126** | **1831** | 2496 |
|  | w/o attn mask | 2534 | 3341 | 3004 | 2956 | 3460 | 2739 | 3593 | 2843 | 3060 | 4238 | 3378 |
|  | w/o ref prompt | 1851 | 2409 | 2093 | 2208 | 2560 | 2192 | 2464 | 1637 | 2571 | 2258 | 2948 |
|  | Ours (2k) | 2035 | 3034 | 2264 | 2594 | 2992 | 2559 | 3373 | 1920 | 2633 | 3753 | 2958 |
|  | Ours (4k) | 1950 | 2541 | 2101 | 2591 | 2261 | 2259 | 2909 | 1660 | 2677 | 2671 | 2495 |
|  | Ours (6k) | 1702 | 2226 | 2114 | 2446 | 2211 | 1985 | 2529 | 1951 | 2451 | 2017 | **2191** |
| Dynamic Degree↑ | Ours | 0.6 | 0.8 | 0.0 | 1.0 | 0.6 | 0.2 | 1.0 | 1.0 | 0.6 | 1.0 | 1.0 |
|  | Ours(one-shot) | **0.6** | **0.8** | 0.4 | **1.0** | 0.6 | 0.4 | **1.0** | **1.0** | **0.6** | **1.0** | **1.0** |
|  | w/o attn mask | 0.6 | **1.0** | **0.6** | 0.8 | **0.8** | 0.8 | 1.0 | 1.0 | 0.2 | 0.4 | 1.0 |
|  | w/o ref prompt | 0.6 | 0.8 | 0.0 | 0.4 | 0.6 | 0.2 | 1.0 | 1.0 | 0.4 | 0.8 | 1.0 |
|  | Ours (2k) | 0.4 | 0.2 | 0.0 | 0.8 | 0.6 | 0.2 | 0.6 | 1.0 | 0.2 | 0.4 | 1.0 |
|  | Ours (4k) | 0.4 | 0.6 | 0.0 | 0.8 | 0.6 | 0.4 | 0.6 | 1.0 | 0.2 | 0.4 | 1.0 |
|  | Ours (6k) | 0.6 | 0.8 | 0.0 | 1.0 | 0.6 | 0.2 | 0.8 | 1.0 | 0.6 | 0.8 | 1.0 |
| EOS↑ | Ours | 1.00 | 1.00 | 1.00 | 1.00 | 1.00 | 1.00 | 1.00 | 1.00 | 1.00 | 1.00 | 1.00 |
|  | Ours(one-shot) | **1.00** | **1.00** | **1.00** | **1.00** | **1.00** | **1.00** | **1.00** | **1.00** | **1.00** | **1.00** | **1.00** |
|  | w/o attn mask | 0.40 | 1.00 | 1.00 | 1.00 | 1.00 | 0.80 | 0.60 | 0.60 | 0.80 | 1.00 | 1.00 |
|  | w/o ref prompt | 1.00 | 1.00 | 1.00 | 1.00 | 1.00 | 1.00 | 1.00 | 1.00 | 1.00 | 1.00 | 1.00 |
|  | Ours (2k) | 1.00 | 1.00 | 1.00 | 1.00 | 1.00 | 1.00 | 1.00 | 1.00 | 1.00 | 1.00 | 1.00 |
|  | Ours (4k) | 1.00 | 1.00 | 1.00 | 1.00 | 1.00 | 1.00 | 1.00 | 1.00 | 1.00 | 1.00 | 1.00 |
|  | Ours (6k) | 1.00 | 1.00 | 1.00 | 1.00 | 1.00 | 1.00 | 1.00 | 1.00 | 1.00 | 1.00 | 1.00 |
| EFS↑ | Ours | 0.0 | 0.6 | 0.2 | 0.6 | 0.8 | 0.6 | 0.0 | 0.6 | 0.6 | 0.8 | 0.4 |
|  | Ours(one-shot) | **0.2** | **0.6** | **0.6** | **0.8** | **1.0** | **1.0** | **0.6** | **1.0** | **0.8** | **0.8** | **0.4** |
|  | w/o attn mask | 0.0 | 0.2 | 0.0 | 0.0 | 0.4 | 0.2 | 0.0 | 0.0 | 0.2 | 0.2 | 0.0 |
|  | w/o ref prompt | 0.0 | 0.6 | 0.2 | 0.4 | 0.8 | 0.6 | 0.0 | 0.2 | 0.6 | 0.6 | 0.4 |
|  | Ours (2k) | 0.0 | 0.2 | 0.2 | 0.6 | 0.8 | 0.6 | 0.0 | 0.2 | 0.6 | 0.4 | 0.4 |
|  | Ours (4k) | 0.0 | 0.4 | 0.2 | 0.4 | 0.8 | 0.6 | 0.0 | 0.4 | 0.8 | 0.6 | 0.4 |
|  | Ours (6k) | 0.0 | 0.6 | 0.2 | 0.4 | 0.8 | 0.6 | 0.0 | 0.4 | 0.6 | 0.8 | 0.4 |
| CLS↑ | Ours | 0.8 | 1.0 | 0.8 | 0.8 | 0.8 | 0.8 | 0.8 | **0.8** | 1.0 | 1.0 | 0.4 |
|  | Ours(one-shot) | **0.8** | **1.0** | **1.0** | **1.0** | **1.0** | **1.0** | **0.8** | 0.6 | **1.0** | **1.0** | **0.6** |
|  | w/o attn mask | 0.2 | 0.4 | 0.0 | 0.2 | 0.4 | 0.2 | 0.0 | 0.0 | 0.4 | 0.2 | 0.0 |
|  | w/o ref prompt | 0.8 | 1.0 | 0.6 | 0.8 | 0.8 | 0.8 | 0.8 | 0.8 | 1.0 | 0.8 | 0.4 |
|  | Ours (2k) | 0.8 | 1.0 | 0.8 | 0.8 | 0.8 | 0.8 | 0.8 | 0.6 | 1.0 | 1.0 | 0.4 |
|  | Ours (4k) | 0.8 | 1.0 | 0.8 | 0.8 | 0.8 | 0.8 | 0.8 | 0.8 | 1.0 | 1.0 | 0.4 |
|  | Ours (6k) | 0.8 | 1.0 | 0.8 | 0.8 | 0.8 | 0.8 | 0.8 | 0.8 | 1.0 | 1.0 | 0.4 |

| Metrics | Methods | Hair | Flight | Illustration | BOOM | Mask | Pizza | Shadow | Spirit_Animal | To_Monkey | Avg. |
|---|---|---|---|---|---|---|---|---|---|---|---|
| FVD↓ | Ours | **2449** | 2960 | 1588 | 2442 | 3101 | 1898 | 1927 | 2664 | 1963 | 2153 |
|  | Ours(one-shot) | 2602 | **2384** | **1330** | **2366** | **3003** | **1841** | **1895** | **2513** | **1889** | **2047** |
|  | w/o attn mask | 4554 | 4158 | 3140 | 3754 | 4650 | 2967 | 3123 | 3601 | 4242 | 3467 |
|  | w/o ref prompt | 3571 | 3163 | 1921 | 3047 | 3498 | 2266 | 2214 | 2664 | 2132 | 2483 |
|  | Ours (2k) | 3837 | 3730 | 2374 | 3457 | 4521 | 2496 | 2379 | 3407 | 2440 | 2938 |
|  | Ours (4k) | 3859 | 2860 | 1904 | 3031 | 4368 | 2173 | 2068 | 2935 | 2126 | 2572 |
|  | Ours (6k) | 2528 | 2935 | 1872 | 3081 | 3736 | 2171 | 2011 | 2807 | 2037 | 2350 |
| Dynamic Degree↑ | Ours | 1.0 | 1.0 | 0.6 | 1.0 | 1.0 | 1.0 | 0.4 | 1.0 | 1.0 | 0.79 |
|  | Ours(one-shot) | **1.0** | **1.0** | **0.6** | **1.0** | **1.0** | **1.0** | **0.8** | **1.0** | **1.0** | **0.84** |
|  | w/o attn mask | 1.0 | 1.0 | 0.6 | 1.0 | 1.0 | 1.0 | 0.4 | 1.0 | 1.0 | 0.81 |
|  | w/o ref prompt | 1.0 | 1.0 | 0.6 | 1.0 | 1.0 | 1.0 | 0.4 | 1.0 | 1.0 | 0.74 |
|  | Ours (2k) | 0.8 | 1.0 | 0.2 | 0.4 | 1.0 | 0.8 | 0.4 | 1.0 | 1.0 | 0.60 |
|  | Ours (4k) | 0.8 | 1.0 | 0.4 | 0.4 | 1.0 | 0.8 | 0.4 | 1.0 | 1.0 | 0.64 |
|  | Ours (6k) | 0.8 | 1.0 | 0.4 | 0.6 | 1.0 | 1.0 | 0.4 | 1.0 | 1.0 | 0.70 |
| EOS↑ | Ours | 1.00 | 1.00 | 1.00 | 1.00 | 1.00 | 1.00 | 1.00 | 1.00 | 1.00 | 1.00 |
|  | Ours(one-shot) | **1.00** | **1.00** | **1.00** | **1.00** | **1.00** | **1.00** | **1.00** | **1.00** | **1.00** | **1.00** |
|  | w/o attn mask | 0.80 | 1.00 | 1.00 | 1.00 | 1.00 | 1.00 | 0.80 | 1.00 | 1.00 | 0.89 |
|  | w/o ref prompt | 1.00 | 1.00 | 1.00 | 1.00 | 1.00 | 1.00 | 1.00 | 1.00 | 1.00 | 1.00 |
|  | Ours (2k) | 1.00 | 1.00 | 1.00 | 1.00 | 1.00 | 1.00 | 0.80 | 1.00 | 1.00 | 0.97 |
|  | Ours (4k) | 1.00 | 1.00 | 1.00 | 1.00 | 1.00 | 1.00 | 1.00 | 1.00 | 1.00 | 0.99 |
|  | Ours (6k) | 1.00 | 1.00 | 1.00 | 1.00 | 1.00 | 1.00 | 1.00 | 1.00 | 1.00 | 1.00 |
| EFS↑ | Ours | 0.8 | 0.6 | 0.0 | 0.2 | 0.0 | 1.0 | 0.4 | 0.8 | 0.4 | 0.47 |
|  | Ours(one-shot) | **1.0** | **0.6** | **0.6** | **0.4** | **0.2** | **1.0** | **1.0** | **0.8** | **0.6** | **0.70** |
|  | w/o attn mask | 0.0 | 0.0 | 0.0 | 0.2 | 0.0 | 0.2 | 0.0 | 0.4 | 0.0 | 0.11 |
|  | w/o ref prompt | 0.8 | 0.6 | 0.0 | 0.0 | 0.0 | 1.0 | 0.2 | 0.8 | 0.2 | 0.40 |
|  | Ours (2k) | 0.4 | 0.4 | 0.0 | 0.0 | 0.0 | 0.8 | 0.4 | 0.6 | 0.2 | 0.34 |
|  | Ours (4k) | 0.8 | 0.4 | 0.2 | 0.0 | 0.0 | 0.8 | 0.4 | 0.6 | 0.2 | 0.40 |
|  | Ours (6k) | 0.8 | 0.6 | 0.0 | 0.0 | 0.0 | 0.6 | 0.6 | 0.8 | 0.2 | 0.42 |
| CLS↑ | Ours | 0.6 | 1.0 | **1.0** | 0.4 | 0.4 | 1.0 | 1.0 | 0.8 | 0.6 | 0.79 |
|  | Ours(one-shot) | **0.8** | **1.0** | 0.8 | **0.6** | **0.4** | **1.0** | **1.0** | **1.0** | **1.0** | **0.87** |
|  | w/o attn mask | 0.2 | 0.6 | 0.4 | 0.0 | 0.0 | 0.8 | 0.4 | 0.0 | 0.4 | 0.24 |
|  | w/o ref prompt | 0.6 | 1.0 | 0.8 | 0.4 | 0.4 | 1.0 | 0.8 | 1.0 | 0.6 | 0.76 |
|  | Ours (2k) | 0.6 | 1.0 | 1.0 | 0.4 | 0.4 | 1.0 | 1.0 | 0.6 | 0.6 | 0.77 |
|  | Ours (4k) | 0.6 | 1.0 | 0.6 | 0.4 | 0.4 | 1.0 | 1.0 | 0.8 | 0.4 | 0.76 |
|  | Ours (6k) | 0.6 | 1.0 | 0.8 | 0.4 | 0.4 | 1.0 | 1.0 | 1.0 | 0.6 | 0.79 |

