# OpenReview forum: "VFXMaster: Unlocking Dynamic Visual Effect Generation via In-Context Learning"
_ICLR.cc/2026/Conference — ICLR 2026 Conference Withdrawn Submission_

### Official Review · Reviewer_umU3 · 2025-10-26

**Soundness:** 3
**Presentation:** 2
**Contribution:** 2
**Rating:** 2
**Confidence:** 5

**Summary:**

This paper presents VFXMaster, a diffusion model that can transfer the visual effect from a reference video to a target video. The core idea is to employ a reference-based in-context learning method, where both reference and target prompt and video sequences are concatenated and fed into a MM-DiT model. An in-context attention mask is used to guarantee the information flow and prevents leakage of non-effect content. In additional, the author also proposes one-shot effect adaptation when the test effect is out of the domain. A small set of newly-inserted concept enhance tokens is fine-tuned on the single reference video while the base model is frozen.

**Strengths:**

1. The framework does not need one lora per effect, increasing the scalability of the model.

2. Strong empirical results are presented. It achieves better performance compared to competitors like VFX Creator and Omini-Effects.

**Weaknesses:**

1. The major concern is the limited novelty of the proposed method. The proposed in-context conditioning for VFX generation is quite straightforward. The example-query in-context learning is already common in the generation field, many works adopts a similar idea (i.e. IP-Adapter, PuLID). The in-context attention mask is also not new.

2. There is no comprehensive studies on the design of attention mask. In ablation, only with and without attention mask results are compared. However, more ablation on different choices of masks are needed.

3. The proposed in-context conditioning can only be applied to MM-DiT framework like CogVideoX. For recent state-of-the-art framework like Wan, which adopts the cross attention mechanism, the proposed method is not applicable. This significantly reduce the generalization ability of the method.

4. Also, adopting such in-context conditioning will significantly increasing the training and inference cost, as the attention window size is significantly enlarged.

**Questions:**

Please see the weaknesses.

---

### Official Review · Reviewer_P4Mo · 2025-10-30

**Soundness:** 2
**Presentation:** 2
**Contribution:** 2
**Rating:** 4
**Confidence:** 4

**Summary:**

This paper introduces VFXMaster, a novel framework for generating dynamic visual effects (VFX) in videos. The core problem it addresses is the poor scalability and lack of generalization of existing methods, which often rely on a "one-LoRA-per-effect" paradigm. VFXMaster reframes VFX generation as a reference-based imitation task, leveraging in-context learning. Given a reference video showcasing a specific effect and a target image, the model can reproduce that effect on the target content.

The main contributions are:

1. A Unified, Reference-Based Framework: VFXMaster is proposed as the first unified model capable of learning a general VFX imitation capability from a diverse dataset, rather than memorizing a closed set of effects.

2. In-Context Conditioning with an Attention Mask: The model is conditioned on a reference prompt-video pair as an "example" and a target prompt-image pair as a "query". A novel in-context attention mask is designed to isolate and inject only the essential effect attributes from the reference, preventing content leakage (e.g., transferring the background or subject from the reference video).

3. Efficient One-Shot Effect Adaptation: To improve generalization to challenging, Out-of-Domain (OOD) effects, the paper proposes a method to fine-tune a small set of "concept-enhance tokens" on a single user-provided video, allowing the model to quickly adapt to new effects with minimal computational cost.

4. A New Dataset and Evaluation Metric: The authors have collected a new dataset of 10k VFX videos and propose a new VLM-based evaluation metric, VFX-Cons., to assess effect generation quality in terms of occurrence, fidelity, and content leakage.

The experiments demonstrate that VFXMaster outperforms existing methods on in-domain effects and shows strong generalization to unseen effects, particularly when augmented with the one-shot adaptation mechanism.

**Strengths:**

1. Novel Problem Formulation: The most significant strength is the shift from specialized, closed-set VFX models to a unified, general-purpose imitation framework. By framing the task as in-context learning, the paper presents an elegant solution to the critical challenges of scalability and generalization that have limited prior work.

2. Effective Architectural Design: The in-context attention mask is a crucial and well-motivated component. The ablation study convincingly demonstrates its necessity, showing a catastrophic drop in performance without it. This highlights that the authors have not just concatenated inputs but have carefully engineered the information flow to achieve the desired outcome (effect transfer without content leakage).

3. Strong Empirical Evaluation: The paper is supported by a thorough and multi-faceted experimental evaluation. The inclusion of quantitative metrics, qualitative comparisons, detailed ablation studies, a data scaling analysis, and a user study provides a robust body of evidence for the method's effectiveness.

4. Valuable Community Resources: The commitment to release the model, code, and the newly curated 10k video dataset is a major strength. This will lower the barrier to entry for other researchers and serve as a benchmark for future work on generalizable VFX generation.

**Weaknesses:**

1. Ambiguity and Potential Flaw in the VFX-Cons. Metric: The paper's new metric, VFX-Cons., is calculated as (EOS + EFS + CLS) / 3. However, the paper states, "CLS is only meaningful when EFS is True." The formula does not reflect this dependency. For example, if a video has the effect occur (EOS=True) but the fidelity is wrong (EFS=False), what is the value of CLS? If it is judged as True (no leakage), the score would be (1 + 0 + 1) / 3 = 0.67. If it is judged as False, the score is (1 + 0 + 0) / 3 = 0.33. The CLS score seems ill-defined when EFS is false, and averaging them masks the individual performance and the logical dependency. This is a significant weakness in a claimed contribution.

2. Fairness of Baseline Comparison: The baseline CogVideoX* is fine-tuned on the same data, but it lacks the reference-based architectural modifications of VFXMaster. The comparison therefore demonstrates that the proposed architecture is better than a generic I2V model for this task, which is expected. A stronger baseline might have involved a simpler method for incorporating reference information into CogVideoX (e.g., via cross-attention to the reference video's features) to better isolate the benefits of the proposed in-context learning paradigm versus simpler reference-based conditioning.

3. Lack of Discussion on Limitations and Failure Cases: The paper presents very successful qualitative results but does not include a discussion on the method's limitations or common failure modes. For which types of effects does it still struggle? Does the one-shot adaptation sometimes lead to overfitting on the single reference video? For instance, does it fail to separate style from motion in complex effects? A dedicated "Limitations" section would make the work more complete and scientifically rigorous.

4. Minor Terminological Point on "One-Shot": The "one-shot effect adaptation" involves fine-tuning on a single video with data augmentations. This creates a mini-dataset from one sample for multiple training steps. While this is an efficient adaptation, the term "one-shot" can sometimes imply inference with a single example without any gradient updates. Clarifying the exact number of training steps and the process would add precision to the "efficient" claim.

**Questions:**

1. Regarding the VFX-Cons. metric: Could you please clarify how the final score is calculated, specifically addressing the dependency of CLS on EFS? The current formula (EOS + EFS + CLS) / 3 seems to ignore this dependency. How is CLS evaluated if EFS is False? Would a hierarchical or weighted scoring system that reflects this dependency be more appropriate? Also, which specific VLM was used for the evaluation (the paper cites a future work, Comanici et al., 2025)?

2. Regarding the baseline comparison: Can you elaborate on why the direct fine-tuning of CogVideoX is considered a sufficient baseline? Did you consider or experiment with any alternative, simpler ways to condition the baseline model on the reference video, which might provide a more direct comparison to your in-context learning approach?

3. Could you discuss the primary failure modes or limitations of VFXMaster? Are there specific categories of visual effects (e.g., those involving complex fluid dynamics, intricate topological changes, or very subtle textural effects) where the model's performance degrades?

4. For the "efficient one-shot effect adaptation," could you provide more detail on the fine-tuning process? Specifically, how many training steps or epochs are performed on the augmented data from the single video? This would help in quantifying the actual computational cost and "efficiency" of the adaptation.

---

### Official Review · Reviewer_CQuf · 2025-10-31

**Soundness:** 2
**Presentation:** 3
**Contribution:** 2
**Rating:** 4
**Confidence:** 4

**Summary:**

This paper solves the problem of visual effect generation. Most previous work mostly adopt the one-LoRA-per-effect framework to generate visual effect. Differently, this paper designs a unified, reference-based pipeline for visual effect video generation. An in-context attention mask is introduced to decouple and inject the essential effect attributes. This enables a single unified model to handle the effect imitation without information leakage. An efficient one-shot effect adaptation mechanism is also proposed to boost the generalization capability of the proposed approach on difficult unseen effects from a single user-provided video rapidly.

Experiments show that the proposed approach receives good results compared to previous works and the authors claim that the code associated with this paper would be made publicly available.

**Strengths:**

- This paper presents a unified reference-based pipeline for visual effect video generation with an in-context attention mask. Compared to previous works, the motivation of this paper is clear. Instead of tuning one lora for each visual effect, this paper aims to handle all visual effects in a single framework, which is meaningful for this research topic.

- The visual results and numerical results look great compared to previous works.

**Weaknesses:**

- In the introduction section, the authors say that most previous VFX generation methods are based on Lora finetuning and they list many references in the second and third paragraphs. However, in the experiment section, it seems that the authors did not compare their approach with these mentioned works. It is hard to say that the proposed approach performs better than these models.

- Presenting a unified model for VFX generation is an interesting work. However, the technical contributions of this paper is not significant enough. The masking strategy has already been used in previous LLM design. Though the proposed masking strategy does work in the proposed framework, I cannot see some new techniques are proposed.

- In the caption of Fig. 2, the authors split it into three parts. However, the authors did not clearly explain which point corresponds to which part in the figure. I think this should be further explained.

- The datasets scaling part in the ablation study looks strange. When we talk about scaling, we often refers to data of different magnitudes. This paper only shows the results based on the 10k dataset. Maybe the title of the paragraph should be changed.

**Questions:**

- The fond size in most figures are really small. It really takes me some time to see the content.

- In addition, the authors are also encouraged to reorganize the figure presentations. They really look too small as I see there are still some spaces that are not used.

---

### Official Review · Reviewer_5k99 · 2025-10-31

**Soundness:** 3
**Presentation:** 2
**Contribution:** 2
**Rating:** 4
**Confidence:** 3

**Summary:**

The paper proposed a reference-based framework for visual effect (VFX) video generation, called VFXMaster. Specifically, by learning from reference effects via in-context learning, VFXMaster integrates diverse effects into a single model. In addition, with a small set of learnable concept-enhance tokens, VFXMaster can deal with Out-of-Domain (OOD) effects. Extensive experiments demonstrate the effectiveness of the proposed method.

**Strengths:**

* The writing is fluent and logically coherent, exhibiting strong readability.
* The proposed method is highly efficient, requiring only a small number of model parameters to be fine-tuned in order to learn various VFX effects.
* Comprehensive qualitative and quantitative experiments demonstrate the effectiveness of the proposed method.

**Weaknesses:**

* The proposed method lacks novelty. For the in-domain training part, it appears to be a straightforward extension of Custom Diffusion to the video domain. The out-of-domain part, on the other hand, resembles a modified version of prompt tuning.
* The ablation study appears somewhat coarse. According to Table 2, the Attention Mask has a significant impact on the performance of VFXMaster. Conducting a single, superficial ablation on the Attention Mask is insufficient. It would be helpful to adjust the Attn Mask shown in Figure 1 or visualize the interactions between different Q and V pairs within the attention weights.
* For the training of the OOD VFX, how does the time cost compare to injecting a new VFX concept from scratch?

**Questions:**

See 'Weaknesses'.
If my questions can be addressed, I would be happy to raise my score.

---

### Note · Authors · 2025-11-12

I have read and agree with the venue's withdrawal policy on behalf of myself and my co-authors.